# Machine Learning-Based Downscaling of Aerosol Size Distributions from a Global Climate Model

Antti Vartiainen<sup>1,2</sup>, Santtu Mikkonen<sup>1,3</sup>, Ville Leinonen<sup>4</sup>, Tuukka Petäjä<sup>5</sup>, Alfred Wiedensohler<sup>6</sup>, Thomas Kühn<sup>7</sup>, and Tuuli Miinalainen<sup>8</sup>

**Correspondence:** Antti Vartiainen (antti.vartiainen@uef.fi)

#### Abstract.

Air pollution, particularly exposure to ultrafine particles (UFPs) with diameters below 100 nm, poses significant health risks, yet their spatial and temporal variability complicates impact assessments. This study explores the potential of machine learning (ML) techniques in enhancing the accuracy of a global aerosol-climate model's outputs through statistical downscaling to better represent observed data at specific sites. Specifically, the study focuses on the particle number size distributions from the global aerosol-climate model ECHAM-HAMMOZ. The coarse horizontal resolution of ECHAM-HAMMOZ (approx. 200 km) makes modeling sub-gridscale phenomena, such as UFP concentrations, highly challenging. Data from three European measurement stations (Helsinki, Leipzig, and Melpitz) were used as target of downscaling, covering nucleation, Aitken, and accumulation particle size ranges during years 2016—2018. Six different ML methods (Random Forest, XGBoost, Neural Networks, Support Vector Machine, Gaussian Process Regression and Generalized Linear Model) were employed, with hyperparameter optimization and feature selection integrated for model improvement. A separate ML model was trained for each of the sites and size ranges. Results showed a notable improvement in prediction accuracy for all particle sizes compared to the original global model outputs, particularly for the accumulation subrange. Challenges remained particularly in downscaling the nucleation subrange, likely due to its high variability and the discrepancy in spatial scale between the climate model representation and the underlying processes. Additionally, the study revealed that the choice of downscaling method requires careful consideration of spatial and temporal dimensions as well as the characteristics of the target variable, as different particle size ranges or variables in other studies may necessitate tailored approaches. The study demonstrates the feasibility of ML-based downscaling for enhancing air quality assessments. This approach could support future epidemiological studies and inform policies on pollutant exposure. Future integration of ML models dynamically into global climate model frameworks could further refine climate predictions and health impact studies.

<sup>&</sup>lt;sup>1</sup>Department of Technical Physics, University of Eastern Finland, Kuopio, 70211, Finland

<sup>&</sup>lt;sup>2</sup>Advanced Computing Facility, CSC – IT Center for Science Ltd, Espoo, 02150, Finland

<sup>&</sup>lt;sup>3</sup>Department of Environmental and Biological Sciences, University of Eastern Finland, Kuopio, 70211, Finland

<sup>&</sup>lt;sup>4</sup>Aerosol Physics Laboratory, Tampere University, Tampere, 33014, Finland

<sup>&</sup>lt;sup>5</sup>Institute for Atmospheric and Earth System Research (INAR) / Physics, Faculty of Science, University of Helsinki, Helsinki, 00014, Finland

<sup>&</sup>lt;sup>6</sup>Leibniz Institute for Tropospheric Research (TROPOS), Leipzig, 04318, Germany

<sup>&</sup>lt;sup>7</sup>Weather and Climate Impact Research, Finnish Meteorological Institute, Helsinki, 00560, Finland

<sup>&</sup>lt;sup>8</sup>Space and Earth Observation Centre, Finnish Meteorological Institute, Helsinki, 00560, Finland

#### 1 Introduction

40

50

Air pollution is considered one of the leading global health risks, in terms of both associated premature deaths and disability (GBD 2019 Risk Factors Collaborators, 2020). Fine particulate matter (PM<sub>2.5</sub>, particle diameter < 2.5  $\mu$ m) has been found to be especially harmful; a recent report from the Global Burden of Disease study identifies it as the most important environmental health risk factor (McDuffie et al., 2021). Although PM<sub>2.5</sub> has undergone extensive study, exposure to and health impacts of smaller particle sizes, such as ultrafine particles (UFPs) with diameters below 100 nm, remain less well understood (Fuzzi et al., 2015; Vogli et al., 2023). Different sized particles contribute to different aspects of the ambient particle concentration — UFPs mainly control the concentrations in terms of number, while coarser particles control the concentrations in terms of mass (PM<sub>2.5</sub>). The size of aerosol particles influences, for example, their deposition in the human respiratory tract and their reactive surface area, which in turn can affect their potential to cause health problems (Kreyling et al., 2004; Schraufnagel, 2020). According to epidemiological and toxicological studies, UFPs can more easily enter the alveoli in the lungs, and from there reach other organs (Kreyling et al., 2004; Schraufnagel, 2020). Compared to larger particles, they can thus potentially contribute to, for example, diabetes (Bai et al., 2018), cancer (Pagano et al., 1996), and ischemic cardiovascular disease (Downward et al., 2018; Li et al., 2017; Ostro et al., 2015) more strongly. However, due to their high spatial and temporal variability, estimating exposure to UFPs is challenging, leading to uncertain or even conflicting conclusions regarding their health impacts (Vachon et al., 2024a; Schraufnagel, 2020). Currently, both the World Health Organization (World Health Organization, 2010) and the European Union (European Council, 2008) provide guidelines on safe exposure limits for PM<sub>2.5</sub> and PM<sub>10</sub> (diameter < 10 μm), but no such limits exist for UFP. Indeed, according to a recent review of the topic (Schraufnagel, 2020), UFPs are, in many ways, "at the frontier of air pollution research".

As the availability of exposure data limits the potential to conduct epidemiological UFP studies (Vachon et al., 2024a), various approaches have been used to gain more information on UFP number concentrations. To study exposure to pollutants, observations from a scarce network of sites have typically been expanded to cover the study area, such as a city, through methods like land use regression (LUR) (Venuta et al., 2024; Amini et al., 2024; Wolf et al., 2017) or interpolation (Jung et al., 2023). Sometimes, more measurements are done in relatively short campaigns to improve the spatial coverage (Vogli et al., 2023; Downward et al., 2018), or satellite-based observations are added as inputs to LUR models to more accurately represent spatial or temporal variability of pollutants (Zani et al., 2020; Jung et al., 2023; Stafoggia et al., 2019). However, most such studies are focused on pollutants other than UFPs (Lin et al., 2022). In recent years, machine learning (ML) has also been a common tool in improving the accuracy of LUR models, often outperforming traditional statistical methods (Vachon et al., 2024b).

While improved spatial characterization of present-day air quality is valuable for understanding its health implications, predicting how air quality may evolve in the future is also important. Actions taken to mitigate climate change might significantly affect the emissions of pollutants or their precursors, thus hindering the prediction ability of LUR models. Furthermore, since these models are purely descriptive and not integrated with physics-based tools, they cannot be used for studying air quality under varying emission scenarios.

Besides the said statistical methods, local-scale air quality is commonly represented using deterministic models that simulate, for example, the emissions, transport, and transformation of pollutants (Sofiev et al., 2006; Johansson et al., 2022). As the inputs of these models can in principle be modified based on the climate change scenario of interest, they could be suitable for long-term air quality prediction. However, many physics-based models simulate only gaseous pollutants such as NO<sub>x</sub> (Pepe et al., 2016) and O<sub>3</sub> (Sharma et al., 2013); aerosols, if supported, may be limited to larger particle sizes (Friberg et al., 2017), omitting UFP. Additionally, running simulations can be computationally expensive and always requires boundary conditions from global climate models, further increasing computational costs. The physics-based air quality models are also not ideal for all sites, as accounting for urban infrastructure or complex terrain requires detailed information about local topography, which is often either unavailable or not accurately captured by local-scale models. (Hinestroza-Ramirez et al., 2023).

55

Compared to air quality models, global-scale climate models generally produce more output variables, potentially also containing size-resolved representations of aerosols. Simulating long-term global changes in aerosol concentrations is possible with climate models, as they incorporate a broader range of atmospheric processes and feedback mechanisms compared to regional climate models. Since global-scale models are already necessary to generate boundary conditions for regional models, using them directly for generating air quality estimates might seem practical. However, for local-scale air quality estimation, the resolution of current climate models is far too coarse, typically ranging from tens to hundreds of kilometers horizontally and tens to hundreds of meters vertically (Turnock et al., 2020). Particularly for UFPs, the challenge arises from their number concentrations being governed by processes such as primary emissions and secondary formation and growth, which occur both in multiple scales. The initial cluster formation occurs in sub-grid spatial scales and it is highly spatially variable (Dada et al., 2023), including a contribution from traffic as well (Rönkkö et al., 2017), while the growth to Aitken and accumulation mode sizes (see Sect. 3) takes place in synoptical scale (Petäjä et al., 2022). All of this makes the particle size distribution of the nucleation mode highly variable in space and time.

An approach known as downscaling can be applied to the low-resolution outputs of global climate models, with the aim of improving their accuracy in the local scale. In this context, the nested approach of initializing regional climate models with boundary conditions from global simulations is called dynamical downscaling (Maraun and Widmann, 2018). Another technique, statistical downscaling, instead aims to find a statistical dependence between coarse-resolution outputs and local observations of the quantity of interest. This dependence can later be used for output correction as a post-processing step. The benefit of statistical downscaling lies in its computational efficiency compared to the computationally much more costly dynamical downscaling (Xu et al., 2020). Most of the literature on downscaling focuses on correcting meteorological variables such as temperature (Li et al., 2020; Goyal et al., 2011; Kim and Villarini, 2024) and precipitation (Xu et al., 2020; Vandal et al., 2017; Sachindra et al., 2018). Some recent studies have applied statistical downscaling to air quality variables (Miinalainen et al., 2023; Gouldsbrough et al., 2024; Ivatt and Evans, 2020) but only a few to UFP number concentrations (Kohl et al., 2023). Although the statistical methods for downscaling have typically been simple bias corrections or linear regressions (Maraun and Widmann, 2018), many downscaling studies from the past few years have instead utilized ML methods with promising results (Xu et al., 2020; Sachindra et al., 2018; Miinalainen et al., 2023; Gouldsbrough et al., 2024). To our knowledge, however, none so far has applied ML methods to UFP downscaling.

In this study, using various ML methods, we have downscaled aerosol particle number size distributions produced by a global aerosol-climate model to better match observations from three measurement stations. We used data from two urban stations (Helsinki, Finland and Leipzig, Germany) and from one rural background measurement station (Melpiz, Germany). The size distribution was represented by three size ranges, the so-called nucleation, Aitken, and accumulation subranges. We have opted to avoid calling these subranges "modes", as the subranges do not exactly match the conventional mode definitions due to limitations in the size resolution of the climate model representation. Based on these subranges, we categorized the simulated and observed daily average particle number concentrations (PNCs). All three subranges overlap with the UFP size range (

**Figure 1.** Measurement site locations marked on the map of north-eastern Europe. M, L, and H refer to the Melpitz, Leipzig, and Helsinki sites, respectively. The grid lines show the coarse output resolution of ECHAM-HAMMOZ. As can be seen, Leipzig and Melpitz are located in the same grid cell. Coordinates have been rounded to one decimal place.

The Melpitz station (51°32′ N 12°54′ E) is situated in Germany, in the southwest of the small town of Torgau (approx. 20 000 inhabitants), immediately west of the village of Melpitz (Hamed et al., 2010). It is classified as a rural background station (Birmili et al., 2016). Particle number size distribution is measured with SMPS (TSI) with size range of 5 nm to 800 nm. All three subranges were also available from the Melpitz station.

The Leipzig station (51°21′ N 12°26′ E) is situated in the city of Leipzig, Germany (approx. 590 000 inhabitants), about 4 km east of the city center (more detailed description in Birmili et al. (2016)). It is classified as an urban background station. Aitken and accumulation subrange number concentrations are available from this site. The measured particle size range was between 10 nm and 800 nm (using DMPS), and for this reason, the number concentration of the nucleation subrange is not available.

# 4 Analysis methods


# 4.1 Downscaling workflow

Statistical downscaling, in this application, refers specifically to finding and utilizing a relationship between the large number of ECHAM-HAMMOZ output variables and the observed particle number concentrations (PNCs) at the three sites with the aid of ML methods. If the ML models can learn this dependence, they no longer need the measurement data to function, but can predict the PNC based purely on the climate simulation. In other words, the outputs of the ECHAM simulation constitute the inputs of the ML models, and the output of the ML models is the number concentration of a specific size subrange. The ML output is also referred to as the target variable, while the input variables are commonly called features. The inputs are properly

**Figure 2.** Daily averages of measured and simulated PNCs per subrange of particle size distribution (rows) at each of the three sites (columns), from 2016 to 2018. The black bars represent missing data. Partitioning of the data into training, validation, and testing subsets is shown by the dashed vertical lines. Acc, Ait, and Nuc refer to the accumulation, Aitken, and nucleation subranges, respectively.

introduced in Sect. 4.2 and listed exhaustively in Table S1. The downscaling is site-specific, that is, the ML models are trained separately for each station and size subrange.

As the first step of the ML pipeline, the dataset consisting of simulated features (i.e., the simulation outputs of ECHAM175 HAMMOZ) and measured PNC was split into three subsets, known as training, validation, and testing sets (see Sect. 4.2).

Next, the number of input variables was reduced through feature selection (Sect. 4.3), the hyperparameters of the models were tuned (Sect. 4.5), and the models were trained, while evaluating the models' performance at each step where necessary. In ML terminology, hyperparameters refer to the tunable parameters of the ML algorithms. This is in contrast to the internal parameters, such as linear model coefficients or neural network weights, which are not controlled by the user but automatically selected by the algorithms. After the optimized, trained models were obtained, further analysis and comparison was done.

On each iteration step of the hyperparameter optimization, an ML model with particular hyperparameter values was fit to the training data, and the trained model was then tested on the validation data to obtain a measure of its goodness-of-fit. The purpose of this common approach is to avoid overfitting the model to the training data, which would impair its generalizability. A separate validation set was used instead of k-fold cross-validation to avoid temporal leakage of information, where future data is used for training and past for testing (Fraga et al., 2023). Rolling variants of k-fold cross-validation that retain the ordering of the data were considered, but initial tests showed poor generalization across folds, possibly due to the seasonality inherent in the data. Thus, k-fold cross-validation was not used in this study.


When the optimization was finished, the combination of hyperparameters leading to the best validation performance was selected, and the ML model with this configuration was retrained on the combined training and validation subsets. Then,

the model was applied to the testing subset to evaluate its performance on completely unseen data. This was done because there could be a slight, optimistic bias in the validation score when the hyperparameters have been selected to be optimal for the validation subset. Because some of the ML methods (RF and the NNs, see Sect. 4.4) utilize randomness as part of their algorithms, and therefore depend on the initialization of the random number generator, the retraining part was repeated with 50 different seed numbers. In the results, the mean performance is shown for these methods. Finally, the SHAP method (see Sect. 4.6) was applied to the trained models to investigate their use of input variables.

# 4.2 Data preparation






For each station, we used ECHAM-HAMMOZ data from only one ECHAM grid cell, which contained the station coordinates and altitude. The ECHAM-HAMMOZ data with a vertical dimension were interpolated to correspond to the station altitude. This was done by utilizing the CDO command line tool (Schulzweida, 2023), and by using the nearest layer to the surface as well as the second lowest layer. Thus, the global scale simulation data were matched with the point measurements. Both the observed and simulated data were then averaged to daily resolution. We selected 93 variables directly from ECHAM-HAMMOZ data as input variables, covering meteorology, aerosol composition and size distribution. Additional variables were created to replace or complement the existing simulated variables. The u and v components of wind were transformed into two cyclic directional components (north-south, wind ns and east-west, wind ew, both varying between -1 and 1) and one variable for the absolute magnitude of the wind vector (wind speed). Finally, time of the year was represented by time ws (wintersummer variability) and time sa (spring-autumn variability). These were also formed cyclically (and thus vary between -1 and 1) to avoid a large difference between the value of the last day of the year and the first of the next year, which occurs when using linear time and can disrupt the ML models. In practice, transforming a variable into two cyclic components was done by applying sinusoidal functions (sine and cosine) to the proportion of the variable's current value relative to its maximum (for example, time  $ws = \cos([day number]/[days in year])$ ). In our naming convention for all four cyclic features, the positive direction is referenced first (i.e., wind ns = 1 means northerly wind and time ws = 1 means New Year). All in all, these changes resulted in 100 input variables to be considered for the ML models (see Table S1). These variables were chosen for their potential relevance for aerosol formation, transport or removal, or because they represent properties of the particle size distribution in the climate model.

The three years of data were split into three subsets used for training, validation, and testing of the ML models. Because of missing data, not all subsets could cover a full year (See Fig. 2). The last year of data (2018) was reserved for testing to prioritize the completeness of the test results, while the rest of the data were split equally between training and validation. The difference in the number of samples between the subsets therefore depended on the measurement site. As atmospheric phenomena often show strong seasonality, it was deemed beneficial to have data from throughout the year in all subsets, even if increasing the size of the training subsets could also have been useful.

As most of the ML methods used in this study benefit from feature normalization, the input data were normalized to zero mean and one standard deviation, computed from the training set. No other preprocessing techniques were applied. For all

parts of the ML procedure that utilize randomness, the random number generator was initialized with an arbitrary seed number (1024858913).

## 225 4.3 Feature selection






Reducing the number of input variables by removing redundant or less impactful ones can improve the performance of ML models, as well as mitigate unnecessary computational costs. Simpler models are also easier to interpret. Therefore, a feature selection scheme was applied before feeding the simulation data into the ML models. A typical way to drop redundant variables is to see if some of the inputs correlate strongly, and only retain non-correlating ones (up to some threshold value). Dependence between each feature and the target variable, on the other hand, can be an indicator of the feature's relevance. Each criterion alone, however, provides only a partial view and could result either in an ineffective selection or the removal of important information. In addition, as feature selection is still often performed manually rather than in a data-driven manner, these issues could be further exacerbated by human error.

In this study, these ideas were combined to take both redundancy and relevance into account through a data-driven approach. First, a threshold for redundancy (hereafter *red\_thresh*) was selected based on the methods described in Sect. 4.5.1. For each feature, the number of high-correlation pairs (correlations larger than *red\_thresh*) that included the feature was counted. The feature participating in the largest number of such pairs was then dropped. This was repeated until no pair of features exceeded the threshold. In case the number of high-correlation cases was equal for two or more features, the magnitudes of the correlations were compared, and the one with the larger sum of correlations was dropped. If only two highly correlated features remain and the sums are thus equal, we have removed the one that appears earlier in the column order of the input data. After this, another threshold was set for relevance (later *rel\_thresh*), and each feature whose correlation with the target fell below this threshold was also dropped. Both relevance and redundancy were measured by Spearman's correlation coefficient to account for nonlinear dependencies.

The two threshold values were optimized along the model-specific hyperparameters for each ML method and dataset to ensure optimal choice of features. Our approach was inspired by the more commonly used minimum redundancy — maximum relevance (mRMR) method (Ding and Peng, 2005), which we also initially compared to both our approach and a more typical correlation-based selection procedure (not presented here). As further motivation for our approach, it is worth noting that mRMR does not offer a mechanism to adjust the relative weighting of relevance and redundancy, which could be beneficial in certain cases.

Naturally, removing any feature during selection entails a trade-off: the model loses some amount of information, even if that information appears redundant or insignificant in the training data. While feature selection is based on observed correlations and redundancy, it is still possible that a removed feature could improve predictions in future scenarios—particularly for out-of-distribution data. To mitigate this risk, we combined feature selection and model-specific hyperparameter optimization using cross-validation on a holdout validation set, allowing the process to account for generalization performance. Additionally, as a robustness check, we repeated the model training using all available input features (i.e., without applying feature selection), to compare performance and ensure that potentially valuable information was not systematically excluded.

## 4.4 Machine learning methods







In this study, the downscaling task is performed using six different statistical methods. Unless mentioned otherwise, all methods were implemented by the Python package *scikit-learn* version 1.1.1 (Pedregosa et al., 2011). The implementations feature a varying number of hyperparameters, some of which were optimized (listed in Tables S2–S8), while others were either left to their default values or given some other constant value. If some non-default constant value was used, it is mentioned in this section. For more detailed information on the effects of the hyperparameters, readers are referred to the documentation of the methods.

Random Forest (RF) and XGBoost (further abbreviated XGB in figures) are based on different approaches to an ensemble of decision trees. RF takes advantage of randomness to reduce the dependence between individual trees in the ensemble, thereby reducing the ensemble's total variance (Breiman, 2001; Hastie et al., 2009). XGBoost, belonging to the class of gradient boosting methods, generally builds smaller trees with less initial variance and aims to reduce the total bias of the ensemble by sequentially adding trees that correct the errors made by the preceding trees (Friedman, 2001). Unlike the other five methods, XGBoost was implemented by the standalone XGBoost library version 2.0.3 for Python (Chen and Guestrin, 2016). The RandomForestRegressor function from scikit-learn was used for RF.

Neural networks (NN), in their most basic form, are made of layers of interconnected nodes that each produce a linear combination of the incoming signals, which is then transformed by a non-linear activation function (Alpaydin, 2014). The first layer is composed of the inputs, while the last layer produces the output(s). The layers between them are referred to as hidden layers. Such simple feedforward NNs are also known as multilayer perceptrons (MLP). The scikit-learn function MLPRegressor was used as our NN, and two different versions were trained separately: one with one hidden layer (NN1), and another with two hidden layers (NN2). These model architectures were also considered distinct from the point of view of our comparison, increasing the number of methods in the results section from six to seven. Before optimization, two hyperparameters were given constant values based on preliminary tests: batch size was set to 32 and solver to "Adam".

Instead of fitting a complex non-linear function to the training data, the Support Vector Machine (SVM) transforms the data into a higher dimension, where it then fits a linear model to it (Cortes and Vapnik, 1995). In practice, this computationally demanding coordinate transformation can be replaced by a kernel operation by choosing suitable basis functions (Alpaydin, 2014). Additionally, if a data point's distance to the fitted hyperplane were smaller than a specified amount, the point would be ignored by the fit. This way, the model's tolerance to minor errors can be controlled (Alpaydin, 2014). A function called SVR from scikit-learn was used to implement the SVM model. Before optimization, the upper limit for solver iterations (*max\_iter*) was set to 10000, as some unsuitable hyperparameter combinations could cause the iteration to become stuck. Concurrently, the SVM's cache size hyperparameter was increased from the default 200 MB to 1000 MB to avoid issues with insufficient memory.

Gaussian Processes (GP) take a Bayesian approach to ML by conditioning a prior distribution, again represented by a kernel function, on the training data. The mean of the resulting posterior process can then be used as a prediction (Rasmussen and Williams, 2005). GaussianProcessRegressor from scikit-learn was used for this study. Most of its hyperparameters were set

before the optimization, leaving *alpha* as the only optimizable parameter. The number of restarts was set to nine (meaning ten runs in total), and *normalize\_y* was set to True, as recommended for zero-mean, unit-variance priors in the documentation. Additionally, *copy\_X\_train* was set to False, as the training inputs are not changed during the optimization and thus do not need to be saved. An RBF (Radial Basis Function) kernel with length scale bounds (1e-10, 1e2) was selected as the covariance function of the GP. The length scale of the kernel is optimized internally by GaussianProcessRegressor, and not as part of the hyperparameter optimization procedure.

The sixth method in the comparison was the Generalized Linear Model (GLM). It generalizes, and improves upon, linear regression by allowing a non-Gaussian error distribution, and enabling a nonlinear relationship between the inputs and the target through a so-called link function (McCullagh and Nelder, 1989). Nevertheless, GLM does not utilize interactions between inputs unless they are explicitly defined, making it considerably simpler compared to the other methods. Because of its relative simplicity, GLM is not always considered a pure ML method. In scikit-learn, GLM is implemented by the TweedieRegressor function.

# 4.5 Hyperparameter optimization

#### **4.5.1** Optimization methods



Finding the hyperparameter values that result in a model configuration with the highest predictive performance can be seen as an optimization problem, where the *objective function* to be optimized takes the hyperparameters as inputs and produces as output some measure of the goodness-of-fit of the corresponding ML model. Each evaluation of the objective therefore involves training an ML model and testing it against observations, which can make a brute force search through hyperparameter combinations extremely slow. To minimize the number of evaluations, the Bayesian optimization (BO) approach aims to approximate the expensive-to-evaluate objective through a surrogate function, such as a Gaussian Process (Brochu et al., 2010). The surrogate function is updated every time a new point is evaluated, and can be used to strategically select the next point either in a region of uncertainty (favoring exploration) or closer to previously found extrema (favoring exploitation). An *acquisition function* determines which points should be evaluated, and can often be tuned to balance the trade-off between exploration and exploitation.

In this study, a BO algorithm from the Python package bayesian-optimization version 1.4.3 was used to search for optimal values of the hyperparameters and feature selection thresholds (Nogueira, 2014). This implementation uses a GP as the surrogate function. The default kernel for the GP is the Matérn kernel, which has a parameter  $\nu$  controlling the smoothness of the sampled functions. Another tunable parameter of the optimizer is the noise level  $\alpha$  of the GP itself. For the acquisition function, the package's default option is the Upper Confidence Bound (UCB) function

$$UCB(x) = \mu(x) + \kappa \sigma(x)$$
 (1)

where  $\kappa$  controls how much weight should be given to the posterior's standard deviation  $\sigma(x)$  relative to its mean  $\mu(x)$  at some point x of the hyperparameter space. That is, a higher  $\kappa$  favors exploration, focusing the search on regions of higher uncer-

tainty. We have generally used the default settings for both the acquisition function and the GP, apart from some customization that is described next.

Many of the hyperparameters in ML models are either integer-valued (e.g. number of estimators in an ensemble) or categorical (e.g. choice of activation function in NNs), while the GP of the BO algorithm utilized in this study only supports optimization of hyperparameters with continuous values. A common solution to this is to take the point suggested by the acquisition function, and either round the hyperparameters to the closest integer or one-hot-encode the categorical ones before evaluating the objective, depending on which one is needed. As demonstrated by Garrido-Merchán and Hernández-Lobato (2020), this approach causes the GP to ignore that an interval around an integer becomes known when one point is evaluated in the interval, as all values in that interval are rounded to the same integer value. This can lead to unnecessary evaluations and thus slow down the iteration. In the worst case, the algorithm can even become stuck on one point. Therefore, the authors propose that the transformation (i.e., rounding and encoding) of the hyperparameters should be done inside the kernel function, so that the acquisition function gains accurate information about the posterior when evaluating a new point. We have applied this approach to the default Matérn kernel, keeping it otherwise unchanged.

As the range of the hyperparameters can be wide and the general location of the optimum can be uncertain, it can be useful to optimize some hyperparameters logarithmically. This is not supported by the BO package by default, but it is easy to implement by transforming, at the beginning of the objective function, the hyperparameter x in question to  $10^x$ , effectively optimizing the value of the exponent. This transformation was applied to many hyperparameters in almost all ML models, and is also indicated differently in Tables S2–S8.

In addition to BO, the optimization of the hyperparameters was also done using a randomized search (RS), which would be expected to perform worse, as long as the BO iteration proceeds properly. As there are multiple parameters to tune for the optimizer itself that can significantly affect its performance (Snoek et al., 2012), a suboptimal selection could potentially make the BO method inferior to a purely random procedure. In our application, where a large number of models are optimized, it would be highly impractical to inspect every model individually to make sure the BO iteration has succeeded, especially with the limited options for visualization available. Visualizing aspects of the optimization process can make it easier to verify that the parameter space has been thoroughly explored and that fitting the GP has been successful. Due to these limitations, both BO and RS were used. While the models and datasets of this study are relatively small and thus a complex method like BO may not lead to major computational gains, the same methodology could be applied in future studies with more computationally intensive problems. These could be, for example, downscaling of a longer or higher resolution time series, training a single model on data from a large number of sites, or using large deep learning models.

# 4.5.2 Optimization procedure







The selected optimization method, either BO or RS, was executed for 300 iterations. In the case of BO, the first 30 points were also sampled randomly to have sufficient data for the acquisition function to operate on. Another case, called "pure BO" in the results section, was run without sampling these initial points. As for the parameters of the optimizer itself, the  $\nu$  of the Matérn kernel was set to 1.5 (making the samples from the GP once differentiable), while the  $\alpha$  of the GP was set to 1e-2 when the

model had categorical hyperparameters, and left to the default 1e-6 otherwise. Three options (1, 2.5 and 10) were tried for the  $\kappa$  parameter of the acquisition function to account for different needs for exploration and exploitation.

In addition to these five cases (RS, pure BO, and BO with three different values of  $\kappa$ ), two more cases were formed by not optimizing the feature selection (FS) parameters as part of either RS or BO (with  $\kappa=2.5$ ). Hence, all 100 input variables were included in the models. These two cases are called "RS & no FS" and "No FS", respectively. It should be noted that without feature selection, the only hyperparameter of the GP model is  $\alpha$ , and therefore it would not make much sense to use BO to optimize it, as it is also based on fitting a GP. In this case, RS was used instead, meaning that "RS & no FS" and "No FS" refer to the same procedure when GP is concerned.

# 365 4.6 Evaluation of model performance





To assess model performance, we employed five complementary evaluation metrics. In addition to the commonly used mean absolute error (MAE), root mean squared error (RMSE), and Pearson's correlation coefficient (r), we also used the coefficient of determination  $(\rho^2)$  and the scaled MAE (sMAE), defined as:

$$\rho^{2}(y,\hat{y}) = 1 - \frac{\sum_{i=1}^{n} (y_{i} - \hat{y}_{i})^{2}}{\sum_{i=1}^{n} (y_{i} - \bar{y})^{2}}$$
(2)

$$\text{sMAE}(y,\hat{y}) = \frac{\text{MAE}}{\bar{y}} = \frac{1}{\bar{y}} \cdot \frac{1}{n} \sum_{i=1}^{n} |y_i - \hat{y}_i|$$
 (3)

Here, y denotes the vector of observed values,  $\hat{y}$  the predicted values, and  $\bar{y}$  their mean.

The coefficient of determination,  $\rho^2$ , measures the proportion of variance in the observations explained by the model. While often referred to as  $R^2$ , we use  $\rho^2$  to emphasize that in non-linear models this value can be negative, unlike in unconstrained linear regression where  $R^2$  is bounded between 0 and 1. A perfect match between predictions and observations yields  $\rho^2 = 1$ .

The scaled MAE (sMAE), applied e.g. in Mikkonen et al. (2020), normalizes MAE by the mean of the observed values. This allows for better comparability across datasets of differing magnitudes, such as the particle size distribution subranges examined here. As the mean PNC is strictly positive, sMAE remains well-defined throughout.

RMSE and MAE are expressed in the same units as the target variable  $(1/\text{cm}^3)$  in this case), with RMSE penalizing large errors more heavily. Pearson's r, in turn, quantifies the association between predicted and observed values and is insensitive to the scale of errors, making it useful for assessing rank consistency.

For model selection during hyperparameter optimization,  $\rho^2$  was used as the primary criterion of goodness-of-fit.

In addition to numerical evaluation, we applied a game-theoretical interpretation method, SHAP (SHapley Additive exPlanations), to assess the influence of individual input features on model predictions (Lundberg and Lee, 2017; Molnar, 2022). SHAP assigns a contribution value to each feature per prediction, indicating the direction and magnitude of its effect relative to the average prediction. Positive (negative) SHAP values imply that the feature increased (decreased) the model's output.

We used the shap Python package (v0.40.0) with the permutation explainer applied to the test data (Lundberg and Lee, 2017). The results were aggregated across all test days and visualized using beeswarm plots, which show both the distribution and strength of feature effects. In these plots, a consistent increase (or decrease) in SHAP values with rising feature values suggests a positive (or negative) association with the target variable.

The SHAP analysis helps identify which input variables the models depend on most and whether these dependencies align with known atmospheric processes. This interpretability is especially valuable when evaluating black-box models such as neural networks or ensemble methods.

## 5 Results and discussion

#### 5.1 Comparison of ML models

In Figure 3, we present a comparison of all seven downscaling methods across the eight datasets. The performance of the methods varied depending on the dataset: all methods were among the best in at least one of the datasets, but most of them also failed in some cases, yielding  $\rho^2$ s close to, or even less than, zero. Only RF and GP showcased stable performance, as they never resulted in a  $\rho^2$  less than 0.1, and were never among the worst performing methods. On average, XGBoost had the highest  $\rho^2$  (0.263), followed by SVM (0.250). XGBoost was also the best method for four out of the eight datasets. It only failed in the nucleation dataset of Helsinki, where it had a lower  $\rho^2$  than any other model (see also Table S4 for hyperparameters, some of which are atypical). However, this dataset turned out to be difficult for all methods, as none of them were able to reach a  $\rho^2$  above 0.15. Generally, the differences between methods were smaller than the differences between datasets, and in many cases, multiple methods were nearly equal in performance. Only some datasets had one method that clearly outperformed the others; this was XGBoost in the nucleation and accumulation datasets of Melpitz and in the Aitken dataset of Helsinki, and GLM in the accumulation dataset of Leipzig. Additionally, XGBoost and SVM were the two best methods for all subranges from Melpitz, indicating some commonality between these datasets.

Overall, other ML methods have a slight advantage over GLM, as its average  $\rho^2$  is the lowest across datasets (0.176). There is, however, strong variance in its performance, as it is among the best methods in both Leipzig's accumulation dataset and Helsinki's nucleation dataset, but among the worst in the six remaining datasets. In three of the six datasets, it is strictly the weakest, and in the other three, only two methods (NN2 and SVM) perform slightly worse. Particularly, the previously mentioned RF and GP were never outperformed by GLM, except in Leipzig's accumulation dataset. Moreover, GLM results in negative  $\rho^2$  (-0.126) in the Aitken dataset of Helsinki, a drastic difference to all other methods.

For RF and the two neural networks, a mean  $\rho^2$  from 50 different initializations is shown in both the table and the graph in Fig. 3. The magnitude of the  $2\sigma$  confidence intervals, given in parentheses, indicates that randomness had a relatively minor effect on the performance of these models, except for those models that performed poorly to begin with. It is also interesting to compare the two variations of the neural network. In all datasets from Helsinki, adding another layer to the neural network was beneficial. The simpler one-layer network yielded better results in all other datasets. This could be linked to a higher complexity in modeling the particle number size distribution in Helsinki compared to the other sites. This complexity may also

Figure 3. Test set performances ( $\rho^2$ ) of the optimized models for all eight datasets. For the methods that are affected by randomness, the  $2\sigma$  confidence intervals computed from 50 different initializations are also shown in the table. The background colors in the table represent the optimization method used. The methods and the abbreviations are explained in Sections 4.5.1 and 4.5.2. In the cases where multiple optimization methods produced the exact same result, the background is left blank. These cases were Helsinki Ait (Pure BO,  $\kappa=1$ , and  $\kappa=2.5$ ) and Helsinki Acc ( $\kappa=1,\kappa=2.5,\kappa=10$ , and RS). The downscaling method(s) that achieved the highest  $\rho^2$  for a given dataset are shown in bold (differences of less than 0.025 are disregarded).

be reflected in the qualities of the optimized models: the three best models developed for Helsinki's subranges utilize all 100 features, while at least some amount of feature selection was beneficial for all of the other datasets' best models (see Sect. 4.3 for a description of feature selection, and Tables S2–S8 for optimization results). Conversely, the accumulation subrange of Leipzig seems to have been a less complex target for downscaling, as the optimal number of features for it was lower than for other datasets, both when considering the best method (GLM, 19 features) and the average of all methods (28 features). In this case, interactions between features were not needed either, as GLM does not utilize those, unlike the other methods. It is of course possible that having access to more training data or an even wider range of input variables would reveal some interactions that were not found by our current procedure. In that sense, the simplicity of the best model might only indicate that something, like outliers in the training data, confused the more complex methods while not affecting the linear model to the same extent.

**Figure 4.** Daily average particle number concentrations per subrange in 2018 (test set), for all three sites. Measurements are shown in blue, ECHAM-HAMMOZ outputs in green, and the results of downscaling by the best model for each dataset (i.e., the bolded cells in Figure 3) in purple. Goodness-of-fit metrics are reported in the top left corners of each figure, first for the downscaling and then, in parentheses, for ECHAM-HAMMOZ.

## 5.2 Downscaling performance

Figure 4 shows the PNC results of the most successful downscaling methods for the test subset (2018) of each of the eight datasets. In all cases, a clear improvement is observed compared to the original subranges simulated by ECHAM-HAMMOZ, both visually and based on the five metrics shown in the figures. XGBoost achieved the highest ρ² for all subranges from Melpitz and the Aitken subrange from Helsinki. Gaussian process regression resulted in the best model for Leipzig's Aitken subrange and Helsinki's nucleation subrange, support vector machine for Helsinki's accumulation subrange, and the general-ized linear model for Leipzig's accumulation subrange. Generally, the downscaling of the accumulation subrange was most

successful, whereas the nucleation subrange seems to have been more difficult to downscale, resulting in relatively low  $\rho^2$ s in both Helsinki and Melpitz. All three downscaled accumulation subranges have higher  $\rho^2$ s, correlations, and sMAEs than any of the other datasets, even though many peaks and troughs are still estimated incorrectly. The downscaling model trained on the accumulation subrange of Melpitz performs best out of the three, producing a  $\rho^2$  of 0.56 and sMAE of 0.24.

The original ECHAM-simulated nucleation subranges differ significantly from the measured ones, likely contributing to the relatively poor performance of the downscaling models for that subrange. In Melpitz, the  $\rho^2$  of the original is lowest among all subranges (-14.53), and the sMAE is highest (3.36). The strongly negative  $\rho^2$  indicates that the large-scale approximation of the size distribution in ECHAM is a poor representation of the nucleation subrange at this site. In wintertime, the simulation represents the measurements reasonably well, but a strong overestimation is apparent from spring to autumn. Similarly in Helsinki, the correlation between the simulated and measured nucleation subranges is almost nonexistent (0.03) before downscaling. The simulation is unable to predict the high peaks in number concentration during spring and early summer, but instead predicts peaks for autumn, when the measured concentrations are relatively low. By downscaling, these differences can be greatly reduced: sMAE drops from 1.01 to 0.56 in Helsinki and from 3.36 to 0.74 in Melpitz, and the previously negligible correlation in Helsinki increases to 0.38. Thus, even though the performance metrics of the downscaled nucleation subrange are worse even when compared to the non-downscaled ECHAM-simulation of the accumulation subrange in Melpitz, the improvements are significant considering the starting point. Additionally, it should be noted that arithmetic means instead of medians were used in the daily averaging to preserve the highly variable nature of the data. Using medians would smooth the time series, which, while possibly improving downscaling results, would also depict the nucleation subrange less realistically.

The representation of new particle formation and nucleation–sized particles is, on many occasions, not sufficient in global climate models (Williamson et al., 2019). This can be due to, for instance, errors in estimating nucleation rates. As a study by Laakso et al. (2022) shows, ECHAM-SALSA tends to favor partitioning of sulfuric acid to the particle phase due to nucleation over condensation, which may lead to overestimation of nucleation subrange particles. Kokkola et al. (2018) compared ECHAM-SALSA number size distributions to observation data, and their results revealed that at some measurement stations, ECHAM-SALSA overestimates the nucleation mode number concentrations. Furthermore, ECHAM-SALSA does not model new particle formation due to nitrates, which may cause differences between modelled and measured nucleation subrange number concentrations. The representation of nucleation-sized aerosols could be enhanced by including a volatility basis set (VBS) scheme (Donahue et al., 2011), which can improve the representation of secondary organic aerosols. In addition to limitations in representing the nucleation mode, other input variables can also contribute to challenges in downscaling. The coarse spatial resolution of global-scale models naturally limits their ability to accurately capture processes other than just new particle formation.

A strong variability can be seen in both the simulated and measured Aitken subranges at the German sites (Melpitz and Leipzig), although the peaks and troughs match poorly. In winter, the simulated concentrations decrease more than they should. The downscaling methods are generally able to bring the concentrations to a more realistic level, but they fail to capture the true variability in the data. Compared to the subranges simulated by ECHAM-HAMMOZ, the downscaled concentrations no longer fluctuate as rapidly, but instead seem to more carefully follow an average level between the peaks and troughs of

the measured time series. This can be acceptable when the main focus of a study is on long-term averages, and not on e.g. maximum daily exposures. Regardless, an improvement compared to the original is seen in all reported metrics. For the Aitken subrange from Helsinki, the original simulated concentration is mostly too low, and has a less drastic summertime variability compared to the German sites. The downscaling by XGBoost fixes the underestimation and brings the variability closer to that of the measurements. Based on the metrics, the results are quite similar to the other Aitken subrange datasets. The increase in the correlation coefficient from 0.12 to 0.53 is largest out of all datasets, and the improvements in sMAE and  $\rho^2$  are also among the largest.

To summarize, downscaling was generally more effective for larger particle sizes than for smaller ones, and for the rural Melpitz site compared to the urban sites. The eight datasets were further examined through statistical tests comparing the means of the training, validation and testing subsets of the measured PNC (not shown). These tests found significant differences between the years for most subranges and sites, amounting to five out of eight cases in total. If the subsets differ substantially, ML models may struggle to generalize from one dataset to another. To potentially reduce the variation between subsets, the temporal dimension of the data could be expanded beyond three years, thereby enlarging each subset. Training the models with more than one year of data, in particular, could enhance generalization performance. Therefore, we recommend collecting more data for future studies, if possible.

We can compare our results to previous studies to place them in a broader context, although no directly comparable studies exist. For example, Ivatt and Evans (2020) trained an XGBoost model to improve the ozone predictions of a chemistry transport model, and achieved an improvement in Pearson's r of 0.36 (from 0.48 to 0.84). This is similar, though in most cases slighty higher, to the improvement achieved by our models. In addition to a different target variable, their higher time resolution and lower spatial resolution (mean of multiple sites) complicate the comparison. XGBoost was also the most successful model in the study by Venuta et al. (2024), which produced spatiotemporal UFP predictions (logarithm of PNC) with a daily time resolution. Their  $\rho^2$  of about 0.72 was significantly higher than ours, though a direct comparison is again difficult due to the smoothing effect of the log-transform combined with data trimming they performed. Additionally, they used observational weather data instead of climate simulations to train the models. As our models can in theory be used to predict far into the future and produce non-transformed estimates of UFP concentrations, the seemingly lower metrics are still competitive, especially considering the substantial improvement over the ECHAM predictions.

#### **5.3** Notes on optimization





In the table of Figure 3, the method of hyperparameter optimization that resulted in the best model is represented by the color of the cell's background. In the cases where the background of the table is white, multiple optimization methods yielded the exact same hyperparameter values and hence also  $\rho^2$ . This means that the hyperparameters resulting in the highest  $\rho^2$  were discovered either during the initial random steps of the iterations (which were now deterministic due to the fixed seed number), or by the convergence of the BO algorithm to the same hyperparameter values during the non-random steps. The latter was the case for the RF model trained on the Aitken data from Helsinki. This makes sense given that one of the equally performing optimization methods was "Pure BO", which didn't utilize random iterations. For the NN2 model trained on Helsinki's accu-

mulation subrange, on the other hand, the optimum was found from among the initially sampled points in all four identically performing cases.

It can be seen that for most models, the BO methods were superior to RS. However, it is surprising that in some cases randomized search (RS) led to higher  $\rho^2$ . The number of iterations for both approaches was the same, and BO searches the parameter space more methodically, so it should have been able to find a better combination of hyperparameter values. In these situations, it is possible that the hyperparameters don't have a clear optimum, and therefore a reasonably good combination can be found randomly. Then, RS could work slightly better than BO purely by chance. Another possibility is that the few alternatives which were tried for the parameters of the optimizer itself (e.g. *kappa*, *alpha*, and *nu*) were suboptimal for that specific model and dataset. Selecting the parameters correctly can be challenging when the number of different models and datasets is large, and when the options for visualization are limited, such as in high-dimensional spaces. Finally, it is possible that the optimization algorithm itself didn't fully work as intended in these cases, or even got stuck without actually converging on a solution, possibly due to the additional complexity in the acquisition function caused by the treatment of discrete-valued and categorical hyperparameters, as mentioned in Nguyen et al. (2020). This problem could be difficult to diagnose in a comprehensive model comparison study, when every result cannot feasibly be individually inspected.

Simultaneously with the model hyperparameters, feature selection was also optimized through two threshold parameters for redundancy and relevance (see Sections 4.3 and 4.5.2). For the GP models trained on the datasets of Helsinki, both "No FS" and "RS & no FS" involved using RS and were thus identical, for reasons discussed in Sect. 4.5.2. It is also interesting to note that if optimized correctly, our feature selection method could have resulted in practically no selection (i.e., full set of features) by setting the thresholds for redundancy and relevance to 0.99 and 0, respectively. Therefore, it should theoretically always be equal or superior to the "No FS" cases where all 100 features were used without any selection procedure. However, this might be further complicated by the effect of an increased number of hyperparameters on the capability of the optimization algorithm to find the optimum.

In conclusion, BO can improve the results of hyperparameter tuning relative to a randomized search, but can be significantly affected by the selection of the BO parameters and therefore requires careful analysis of the optimization process. Due to this tradeoff between the simplicity of RS and the (generally) improved optimization performance of BO, BO may be preferable when developing one computationally expensive ML model. However, when the number of models under optimization is large, the interpretability and ease of implementation of RS can make it a more practical choice. Other Python packages that implement similar optimization methods, though not only Bayesian, also exist and could alternatively be utilized. Some examples are Hyperopt (Bergstra et al., 2013), Optuna (Akiba et al., 2019), and SMAC3 (Lindauer et al., 2022).

# 5.4 Interpreting the models





The SHAP method, described in Sect. 4.6, was used to analyze the features in the ML models. Figure 5 shows a summary of the most important features across all models. The height of the bars relates to how many models were strongly influenced by the corresponding feature, defined by the feature being among the ten highest when ranked by mean absolute SHAP value. For example, the north-south directional component of wind was among the ten most important in 46 models out of the total 56.

**Figure 5.** Most important input variables across all eight datasets, measured by mean absolute SHAP values. All seven ML models were analyzed for each dataset. Hence, the upper limit for the height of the bars is 56. Bars with height less than ten are not shown.

In general, the wind-related features are seen to be important for the prediction of all subranges of the size distribution, though less so for the nucleation subrange. Solar radiation is also one of the most important variables. ML models for the smallest two size ranges seem to utilize emissions of organic carbon, whereas accumulation subrange is connected to sulfur dioxide  $(SO_2)$  and sulfate  $(SO_4)$ , according to the SHAP values. Interestingly, the feature for geopotential height is mainly used by the Helsinki models.

The modewise summaries of the SHAP explanations (given in Figures S1–S3) can be examined for additional insights. Figure S1 shows that also variables related to dust and black carbon, which are not present in the summary figure (Fig. 5), are contributing to many of the ML models for accumulation subrange. Variables used for downscaling Aitken subrange (Fig. S2) do not substantially differ from the ones shown in the summary of Fig. 5. In general, we recommend refraining from using SHAP to interpret weakly performing models, such as most of the ones for nucleation subrange (Fig. S3), as any conclusions made are likely to be misleading.




The best-performing ML models were studied in detail using SHAP (Figures S4–S15). These model explanations can be compared to experimental studies from the sites to see how well the statistical relationships found by the models correspond to the physical characteristics of the locations. The Helsinki station can be taken as an example. In previous research (Järvi et al., 2009), the surrounding area has been subdivided into three distinct land use sectors, of which the road sector to the southeast has been found to be the largest contributor to the accumulation mode (100–1000 nm), especially during springtime. In addition to the road itself, long-range transport from the east is hypothesized to contribute to this sector's accumulation mode. On the other hand, an increased concentration of ultrafine particles (3–100 nm) has been associated roughly equally with the road sector and the urban sector to the north. The vegetation sector to the west remains a direction of slightly less

polluted air throughout the year. These findings are in line with the effects of wind direction in our models: Figures S4–S8 for Helsinki show that the east-west wind component is important in all models, and that its effect is positive (i.e., easterly wind is connected to increased pollutant concentrations). The two best models for Helsinki's accumulation subrange (Figures S4–S5) both also include the north-south component, which has a negative effect on PNC. This means that the models predict higher concentrations when wind is blowing from the south. Järvi et al. (2009) point out that ship emissions from the harbor, located approximately in this direction, can affect accumulation mode PNC. Moreover, the springtime increase is also captured by these two models. It is interesting to note that NN2 has almost the same  $\rho^2$  as SVM, despite using far fewer features (18 and 100, respectively; see Table S9).

As the models for Helsinki's nucleation subrange (Figures S7–S8) are quite weak, and therefore unlikely to capture the relevant effects, we compare the UFP of Järvi et al. (2009) only to our Aitken subrange downscaling model (Fig. S6). The positive effect of northerly wind and the negative effect of temperature on the Aitken subrange PNC seem realistic, as wood combustion in the urban sector is a significant source of pollutants in the area. Importance of the variables boundary layer height and atmospheric pressure might also be related to the same phenomenon.

Likely, some (or even most) of the features shown in the SHAP plots are only deemed important in terms of their contribution to the downscaling because they correlate with some physically relevant quantity, and not because they themselves cause changes in PNC. For example, this is probably the case with the sea salt variable in Figures S13 and S15, as Melpitz is located nowhere near marine environments. In the ECHAM-HAMMOZ data, PM25\_SS is highly correlated with certain variables (num\_2a6, num\_2a7, WAT\_2a6, and WAT\_2a7) that might more realistically be connected to PNC in Melpitz, however. Hence, when employing SHAP values to assess feature importance, it is important to note that SHAP explains how specific models operate and is not to be interpreted as a tool for causal inference of physical systems.

In this analysis, it should naturally be recognized that all features originate from a simulation of large-scale climate, and therefore do not necessarily represent the immediate surroundings of the measurement sites. Additionally, SHAP is known to be sensitive to correlated features (Aas et al., 2021), which most of our models include; if accurate explanations of the models are crucial, care should be taken to remove all (even somewhat) correlating features before training or use more robust explanation methods.

#### 5.5 Performance considerations







Although our results indicate that some of the ML methods may on average result in higher goodness-of-fit metrics, there are other aspects that might affect the choice of downscaling method. For example, the training and inference durations for different ML methods can vary differently as a function of the dimensionality of input data. In our study, the computational performance of the methods was not considered important, as the downscaling was done as a post-processing step. However, if the downscaling was included as an online correction in the climate model itself, speed of the method would be critical. We have compared the computational performance of the seven model architectures separately for the training, optimization, and inference steps (Fig. S16). There is significant variation in performance: in terms of training, GLM and SVM are by far the fastest, while the NN architectures take longest to train on average. When applying the models for inference, however, NNs

are among the fastest, along with GLM. This might make them preferable in applications where computing time is costly. In the optimization phase, the BO algorithm itself takes relatively long to iterate through, reducing the difference in total duration between most methods; still, NNs are the slowest, though there was large variation depending on the number of NN nodes. Adding a layer to the NNs slowed their training substantially. These findings are naturally only indicative of how the methods perform computationally, and may not apply to datasets of different size. Moreover, parallel computation of the training or inference algorithms can yield additional speedups, which could be another advantage of the methods capable of being parallelized. Of our six methods, RF and NN training can be run in parallel, as the trees in RF can be trained independently, and the NN training can be split into independent batches. Parts of the XGBoost algorithm can also be parallelized, though the trees of the ensemble must still be trained successively (Chen and Guestrin, 2016). Another advantage of NNs is that their structure is ideal for multi-target regression, i.e., the number of target variables can be freely chosen. This way, all three subranges of the size distribution could be downscaled with a single NN. Using the other ML methods, a separate model needs to be trained for each individual output variable.

# 6 Conclusions





This study provides a proof of concept for using ML methods to improve the site-specific accuracy of aerosol particle number size distributions derived from global-scale climate models. By employing six ML methods, optimized through feature selection and hyperparameter tuning, significant improvements were observed in the simulated particle concentrations, especially for accumulation and Aitken subranges. Among the methods, XGBoost demonstrated, on average, superior performance across various datasets. Despite these advances, the nucleation subrange proved more challenging to downscale, likely due to high spatial variability and limitations in the underlying large-scale climate model outputs, particularly in the processes contributing to new particle formation.

The findings underscore the potential of ML-enhanced downscaling as a computationally efficient alternative to traditional methods, offering robust applications in air quality and epidemiological studies. It was observed that downscaling methods can significantly enhance model accuracy at individual measurement sites. However, the selection of a suitable downscaling method requires precision and depends on the target variable's characteristics, as well as spatial and, assumably, temporal dimensions. For example, while particle size ranges were the focus here, the same methods could be applied to other variables as well. Future research should focus on expanding the geographical scope of measurement data, integrating additional features to capture local-scale variations, and exploring dynamic downscaling during climate simulations. Additionally, deep learning methods specialized for time series regression, such as Long Short-Term Memory NNs (Hochreiter and Schmidhuber, 1997), might further improve the downscaling performance by accounting for temporal dependencies in the observed and simulated data (if training data with no missing values were available). These advancements could enhance the predictive accuracy of particle size distributions in coarse-scale climate models, contributing to better assessments of climate change impacts and health outcomes.

*Financial support.* This research has been funded by the UEF Doctoral school and supported by following Research Council of Finland (RCoF) grants: Competitive funding to strengthen university research profiles (PROFI) for the University of Eastern Finland (grant nos. 325022 and 352968), The Atmosphere and Climate Competence Center (ACCC) Flagship (grant nos. 337549, 357902, 359340, 337550, 357904, 359341, 359342, and 359343), Flagship of Advanced Mathematics for Sensing Imaging and Modelling (grant no. 359196), RESE-MON project (grant nos. 330165 and 337552), and ClimAirPathways (grant no. 355531). Additionally, financial support from University of Helsinki via ACTRIS-HY and European Commission via RI-URBANS (101036245) is gratefully acknowledged.

- Code and data availability. The ECHAM6-HAMMOZ model is made available to the scientific community under the HAMMOZ Software License Agreement, which defines the conditions under which the model can be used. The license can be retrieved from https://redmine.hammoz.ethz.ch/attachments/291/License\_ECHAM-HAMMOZ\_June2012.pdf (last access: 17 September 2025). The model data can be reproduced using ECHAM-HAMMOZ model revision 6588 from the repository https://redmine.hammoz.ethz.ch/projects/hammoz/ (HAMMOZ consortium, 2025).
- The Python code files for data processing, machine learning, and plotting are provided in the Zenodo archive at https://doi.org/10.5281/zenodo.17079843 (Vartiainen, 2025). The same archive also contains the ECHAM-HAMMOZ simulation settings, the trained ML models, as well as the processed simulation and measurement datasets.

The particle number size distribution measurement data are openly available in the EBAS (https://ebas.nilu.no/, last access: 17 September 2025) and SmartSMEAR (https://smear.avaa.csc.fi/, last access: 17 September 2025) databases.

Author contributions. AV: conceptualization; methodology; investigation; validation; formal analysis; writing - original draft; visualization. SM: conceptualization; methodology; supervision; writing - review and editing. VL: methodology; investigation; writing - review and editing. TP: investigation; writing - review and editing; data curation. AW: investigation; writing - review and editing; data curation. TK: conceptualization; writing - review and editing. TM: conceptualization; methodology; supervision; writing - review and editing.

Competing interests. One author is a member of the editorial board of journal "Atmospheric Measurement Techniques"

Acknowledgements. We are grateful to Dr Leena Järvi from University of Helsinki (INAR / Physics & HELSUS) for administration of the SMEAR III station in Helsinki.

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
