# Peer review of "Machine Learning-Based Downscaling of Aerosol Size Distributions from a Global Climate Model"

_EGUsphere, 2025_

## Author Comment (AC1)

We gratefully thank both reviewers for their thorough effort in reading the manuscript and providing valuable comments. Point-by-point responses to the reviewers' questions are given below, color-coded for ease of reading. Text in **black** constitutes the original comments by the reviewers, whereas our responses are marked **red,** and suggested changes to the manuscript indicated in **blue**. **When page, section, figure, and line numbers are referenced, we refer to the original, old version of the manuscript.**

In addition to the changes suggested by the reviewers, we noticed we had used a shorter size range than intended when forming the accumulation subrange from the SALSA bins. The upper limit of the SALSA bin 2a3 is 362 nm, not 700 nm, which would instead correspond to bin 2a4 (see Table 1 in Bergman *et al.* (2012)). We extended the range to include bins 2a4 and 2b4, which only had a minor effect on the ECHAM metrics in Figure 3 (in parentheses), as well as the green lines in Figures 2 and 3 (though the difference is too small to notice). This is because only a small fraction of the simulated accumulation subrange particles reside in the 362 – 700 nm range, compared to the 50 – 362 nm range. The main downscaling results were not affected, as the subranges formed from SALSA bins were only used for comparison and not for model training.

Additionally, we have made a few minor changes to the wording of some sentences in the manuscript, as well as some technical corrections to the list of references.

**Reply to Anonymous referee #1**

This work explores the potential of machine learning techniques in enhancing the accuracy of a global aerosol-climate model's outputs through statistical downscaling. The study focuses on the particle number size distributions from ECHAM-HAMMOZ and data from three European measurement stations were used for downscaling. The results show an improvement in prediction accuracy compared to the original global model outputs. It is a complex and extended study and the results fit within the scope of AMT, being of interest for the international research community. However, I would suggest some aspects to be considered in order to improve the manuscript and/or strengthen its impact before it is published in AMT.

We thank the reviewer for these positive comments regarding our manuscript, and for the suggestions given for improvement.

General – The authors state that the ML methods would certainly improve the spatial accuracy of PNSD derived from models. However, in this study, the ML methods are only applied to specific measurement sites. The models are trained using measurements and global model data, but the horizontal resolution of the climate model is 1.9° × 1.9°, which corresponds to approximately 150 km. So the question is: how would the methods perform at other locations within the same grid cell where no measurement data are available for training? For example, in both Helsinki and Leipzig, there are at least two stations measuring PNSD (urban and traffic). I suggest comparing the model performance at two nearby sites, or at least clarifying the intended utility and applicability of the methods and results presented. From the reviewer's perspective, based on the results shown in this manuscript, these methods do not necessarily improve the spatial accuracy of the model but rather enhance the model's ability to reproduce observations at specific measurement locations.

We are thankful to the reviewer for pointing out to us that the wording of the manuscript could indicate to the reader that our downscaling models increase the resolution of the global-scale model on a more general level; it would be reasonable to assume so, as this is what downscaling methods often aim to do. In our study, corrections to the predicted PNC were indeed only done for each site separately, which we have considered a form of downscaling from the large grid cell area to the point represented by a measurement site. As opposed to a large-scale resolution improvement, the intent of our study is to demonstrate the ML-based statistical downscaling methodology on single stations, which could be extended on in the future by training on more stations and e.g. interpolating between them. This is not to say that the results cannot be valuable without future extension – our methodology can be applied to any site of interest (given sufficient PNSD data) to study the evolution of UFP concentrations in e.g. different climate change scenarios or in the past when observations were not yet available. To clarify the applicability of our results, we have done the following modifications to the manuscript:
L5: changed the sentence to "This study explores the potential of machine learning (ML) techniques in enhancing the accuracy of a global aerosol-climate model's outputs through statistical downscaling to better represent observed data **at specific sites**".
L9: added sentence "A separate ML model was trained for each of the sites and size ranges."
L536: changed the sentence to "This study provides a proof of concept for using ML methods to improve the **site-specific** accuracy of aerosol particle number size distributions derived from global-scale climate models."

Furthermore, the reviewer raised concerns over the generalizability of our ML models to other sites than the ones they have been trained on. We agree it is unlikely that the models would work very well at other sites without further training, as even relatively nearby sites could differ greatly in terms of the features affecting the local air quality (e.g., direction of emission sources, topography of buildings, etc.). However, it was not our intention to apply the trained ML models to a larger area surrounding the stations, but rather apply the correction station-wise, as explained above. Therefore, we deemed further analysis of spatial generalizability not relevant for our intended application.

L130 – The authors defined the nucleation, Aitken, and accumulation mode size ranges using uncommon values (<7.7 nm, 7.7–50 nm, and 50–700 nm). This choice appears to be driven by the bin structure of the SALSA model. If that is the case, I would suggest either avoiding the use of the terms "nucleation," "Aitken," and "accumulation" throughout the manuscript, OR adjusting the size ranges to align with the commonly accepted definitions associated with different aerosol processes (e.g., <25 nm, 25–100 nm, and 100–1000 nm). I would also suggest rephrasing: "These size ranges correspond to the SALSA bins…" by "These size ranges were selected to correspond to the SALSA bins…"

We thank the reviewer for the suggestion; the reviewer is right that the size ranges here differ slightly from the typical size ranges used in the aerosol measurement community. We decided to alter the current naming of the size modes, and in addition, clarify the definition as suggested by the reviewer. We have altered the text starting from line 131 to "These size ranges **were selected to** correspond to the SALSA bins 1a1 for nucleation, 1a2–1a3 for Aitken, and 2a1–2a4 for accumulation"

In terms of the naming scheme, we have now aimed to avoid the word "mode" in order to ensure that our definition does not get as easily confused with the conventional definition of the modes. Instead, we have opted to use the term "subrange" (or, in some cases, other similar words, such as "range", "sizes", or "size range"). This change has been applied to all instances of the word "mode" in the manuscript. The only exception is when other studies dealing with modes are referenced. Additionally, when the term is first used on L88, we have added the following clarification:
"We have opted to avoid calling these subranges "modes", as the subranges do not exactly match the conventional mode definitions due to limitations in the size resolution of the climate model representation."

Introduction – The first paragraph of the introduction is unclear regarding the distinction between particle number concentrations and mass concentrations. The two terms appear to be used interchangeably or without clear differentiation. I recommend that the authors clarify when they are referring to number concentrations versus mass concentrations and ensure consistent use of these terms throughout the paragraph. It is important to remain that while UFPs mainly control ambient particle concentrations in terms of number, coarser particles control particle concentrations in terms of mass (PM10 and PM2.5).

We agree with the reviewer that discussing both UFPs and PM2.5 (as well as PM10) in the same paragraph can be misleading, as UFPs are much less relevant when considering the mass concentration. Following the reviewer's suggestion, we have added the following disclaimer on L25:

"Different sized particles contribute to different aspects of the ambient particle concentration – UFPs mainly control the concentrations in terms of number, while coarser particles control the concentrations in terms of mass ($PM_{2.5}$)."

Additionally, we have specified in the introduction whenever we are speaking about number concentrations.

L123-129 – Are the PNSDs measured in Germany obtained using a DMPS or a scanning instrument?

We had used an older reference (Hamed *et al.*, 2010) for the Melpitz station, which still listed a DMPS as the measurement instrument. The reviewer is correct in questioning this – a newer publication (Birmili *et al*., 2016), already used as reference for the Leipzig station, also mentions that the Melpitz station utilizes an SMPS instrument. We have therefore modified the manuscript as follows:

L126: changed "DMPS device" to "DMPS/SMPS (Differential/Scanning Mobility Particle Sizer) instruments"

L143: added the sentence "Particle number size distribution is measured with SMPS (TSI) with size range of 5 nm to 800 nm."

What size ranges does each instrument cover? The comparison with the model would be site-dependent if the size distributions differ in their lower and upper diameter limits. Uncertainties of the measurements are not considered?

We have added information concerning the size ranges and names of the instruments used at each site. Melpitz was already mentioned in response to the above comment; for Helsinki and Leipzig, the following changes were made:

L139: added the sentence "In Helsinki, the particle number size distribution is measured with DMPS (TSI), with size range of 3 nm to 1 μm."

L146: modified "At the Leipzig station, the particle size distribution started at 10 nm, …"
to
"The measured particle size range was between 10 nm and 800 nm (using DMPS)"

As the reviewer mentioned, the differences in size distributions might affect the results of the fitting process. This, besides other factors, such as the representativeness of the model grid cell for the measuring station, suggests that the modeling process should be done separately for each site, i.e. the process is site-dependent. In this study, the focus was on verifying how accurate estimates of the number concentrations of different UFP size ranges can be obtained with current data from climate model runs. We didn't consider the uncertainty related to the measurement data, i.e. the number concentrations reported from the sites were used as "the ground truth" concentrations in training, validation, and test datasets.

Structure – I suggest reconsidering the structure of the sections. For example, the results of the best-performing method are presented in Section 5.1 before the performance of all methods is discussed in Section 5.2. It may be more logical to first present the comparison across all methods, followed by a deeper look at the best-performing one.

Both reviewers suggested this change, and we agree that the revised order is more logical. Accordingly, we have reordered the sections in question.

Additionally, the title of Section 2, "Climate simulation," may not be the most appropriate for the modelling setup. I would suggest something more descriptive, such as "Global model simulations".
We thank the reviewer for the suggestion. On further thought, we realize that "Climate simulation" may be misunderstood e.g. as a longer simulation than the three years we have used. We have renamed the section "Global aerosol-climate model simulation" to make the title more specific.

Nucleation range differences – In several instances, the authors suggest or conclude that "the nucleation mode proved more challenging to downscale due to high spatial variability and limitations in the underlying large-scale climate model output". From the reviewer's perspective, the Aitken mode could also exhibit substantial variability, particularly due to urban emissions. Therefore, a more plausible explanation for the difficulty in downscaling the nucleation mode may lie in the limitations of global models in representing new particle formation (such as the treatment of organics, nitrates, sulfuric acid, or nucleation schemes) rather than primarily in the spatial variability of the sources.

This is an important point.  We agree that global climate models do not always represent new particle formation sufficiently. In addition, as Figure 3 indicates, ECHAM-SALSA may potentially have too strong NPF under certain atmospheric conditions, which can lead to overestimation of the number of nucleation subrange particles. We have therefore added to line 380 the following:

"The representation of new particle formation and nucleation–sized particles is, on many occasions, not sufficient in global climate models (Williamson *et al.*, 2019). This can be due to, for instance, errors in estimating nucleation rates. As a study by Laakso *et al.* (2022) shows, ECHAM-SALSA tends to favor partitioning of sulfuric acid to the particle phase due to nucleation over condensation, which may lead to overestimation of nucleation subrange particles. Kokkola *et al.* (2018) compared ECHAM-SALSA number size distributions to observation data, and their results revealed that at some measurement stations, ECHAM-SALSA overestimates the nucleation mode number concentrations. Furthermore, ECHAM-SALSA does not model new particle formation due to nitrates, which may cause differences between modelled and measured nucleation subrange number concentrations. The representation of nucleation-sized aerosols could be enhanced by including a volatility basis set (VBS) scheme (Donahue *et al.*, 2011), which can improve the representation of secondary organic aerosols."

Technical corrections

L275 – what means the "-" at the end of the reference?
We removed the dash (which was included in the citation given on the GitHub page) and also added the date of last access.

L130-131 – change "These size ranges correspond to the SALSA bins…" by "These size ranges were selected to correspond to the SALSA bins…"
As mentioned previously, we have now made this change.

L280 - should $\sigma(\cdot)$ and $\mu(\cdot)$ be $\sigma(x)$ and $\mu(x)$? Actually "x" (eq. 1) is not defined.
We thank the reviewer for noticing this discrepancy. We have removed the placeholder dots and defined the argument "x" of the UCB function on L280:
"…where κ controls how much weight should be given to the posterior's standard deviation **$\sigma(x)$** relative to its mean **$\mu(x)$ at some point x of the hyperparameter space**"

References:

Bergman, T. ,Kerminen, V.-M., Korhonen, H., Lehtinen, K. J., Makkonen, R., Arola, A., Mielonen, T., Romakkaniemi, S., Kulmala, M., and Kokkola, H.: Evaluation of the sectional aerosol microphysics module SALSA implementation in ECHAM5-HAM aerosol-climate model., Geosci. Model Dev., 5, 845–868, https://doi.org/10.5194/gmd-5-845-2012, 2012.

Birmili, W., Weinhold, K., Rasch, F., Sonntag, A., Sun, J., Merkel, M., Wiedensohler, A., Bastian, S., Schladitz, A., Löschau, G., Cyrys, J., Pitz, M., Gu, J., Kusch, T., Flentje, H., Quass, U., Kaminski, H., Kuhlbusch, T. A. J., Meinhardt, F., Schwerin, A., Bath, O., Ries, L., Gerwig, H., Wirtz, K., and Fiebig, M.: Long-term observations of tropospheric particle number size distributions and equivalent black carbon mass concentrations in the German Ultrafine Aerosol Network (GUAN), Earth System Science Data, 8, 355–382, https://doi.org/10.5194/essd-8-355-2016, 2016.

Donahue, N. M., Epstein, S. A., Pandis, S. N., and Robinson, A. L.: A two-dimensional volatility basis set: 1. organic-aerosol mixing thermodynamics, Atmos. Chem. Phys., 11, 3303–3318, https://doi.org/10.5194/acp-11-3303-2011, 2011.

Hamed, A., Birmili, W., Joutsensaari, J., Mikkonen, S., Asmi, A., Wehner, B., Spindler, G., Jaatinen, A., Wiedensohler, A., Korhonen, H., Lehtinen, K. E. J., and Laaksonen, A.: Changes in the production rate of secondary aerosol particles in Central Europe in view of decreasing SO2 emissions between 1996 and 2006, Atmospheric Chemistry and Physics, 10, 1071–1091, https://doi.org/10.5194/acp-10-1071-2010, 2010.

Kokkola, H., Kühn, T., Laakso, A., Bergman, T., Lehtinen, K. E. J., Mielonen, T., Arola, A., Stadtler, S., Korhonen, H., Ferrachat, S., Lohmann, U., Neubauer, D., Tegen, I., Siegenthaler-Le Drian, C., Schultz, M. G., Bey, I., Stier, P., Daskalakis, N., Heald, C. L., and Romakkaniemi, S.: SALSA2.0: The sectional aerosol module of the aerosol–chemistry–climate model ECHAM6.3.0-HAM2.3-MOZ1.0, Geosci. Model Dev., 11, 3833–3863, https://doi.org/10.5194/gmd-11-3833-2018, 2018.

Laakso, A., Niemeier, U., Visioni, D., Tilmes, S., and Kokkola, H.: Dependency of the impacts of geoengineering on the stratospheric sulfur injection strategy – Part 1: Intercomparison of modal and sectional aerosol modules, Atmos. Chem. Phys., 22, 93–118, https://doi.org/10.5194/acp-22-93-2022, 2022.

Williamson, C.J., Kupc, A., Axisa, D. *et al*: A large source of cloud condensation nuclei from new particle formation in the tropics, *Nature,* **574**, 399–403, https://doi.org/10.1038/s41586-019-1638-9, 2019.

---

## Author Comment (AC2)

We gratefully thank both reviewers for their thorough effort in reading the manuscript and providing valuable comments. Point-by-point responses to the reviewers' questions are given below, color-coded for ease of reading. Text in **black** constitutes the original comments by the reviewers, whereas our responses are marked **red,** and suggested changes to the manuscript indicated in **blue**. **When page, section, figure, and line numbers are referenced, we refer to the original, old version of the manuscript.**

In addition to the changes suggested by the reviewers, we noticed we had used a shorter size range than intended when forming the accumulation subrange from the SALSA bins. The upper limit of the SALSA bin 2a3 is 362 nm, not 700 nm, which would instead correspond to bin 2a4 (see Table 1 in Bergman *et al.* (2012)). We extended the range to include bins 2a4 and 2b4, which only had a minor effect on the ECHAM metrics in Figure 3 (in parentheses), as well as the green lines in Figures 2 and 3 (though the difference is too small to notice). This is because only a small fraction of the simulated accumulation subrange particles reside in the 362 – 700 nm range, compared to the 50 – 362 nm range. The main downscaling results were not affected, as the subranges formed from SALSA bins were only used for comparison and not for model training.

Additionally, we have made a few minor changes to the wording of some sentences in the manuscript, as well as some technical corrections to the list of references.

**Reply to Anonymous referee #2**

This manuscript presents an interesting application of machine learning (ML) methods for downscaling particle number size distributions from a global climate model. The topic is relevant for improving exposure estimates in air quality studies and refining global climate model outputs. The authors provide a detailed description of the methods and an extensive discussion of the results. However, the manuscript would benefit from significant revisions to improve its clarity in the methodology and results. In particular, several aspects of the study design, including the choice and justification of ML methods, the feature selection strategy and the interpretation of SHAP results, require clearer explanations and additional justification. In its current form, I believe the manuscript requires major revisions before it can be considered for publication.

We thank the reviewer for the positive comments and constructive feedback regarding our manuscript.

**General comments:**

1. In Section 2, the description of the climate simulation with the global climate model would benefit from a clearer explanation of which model outputs are compared with aerosol measurements. The text mentions that the SALSA 2.0 module discretizes the aerosol size distribution into ten size classes, but it is not entirely clear how these classes are defined and whether any conversions or postprocessing steps are applied before comparison with the measurements. While some of this information is briefly introduced later in Section 3 (lines 130–132), it would improve clarity to include this explanation earlier in Section 2, when the SALSA module is first introduced.

We agree with the reviewer that explaining the purpose of the model outputs would clarify the manuscript. Therefore, we have added the following sentences on line 109:
"The size classes range from 3 nm to 10 μm, from which we have selected the seven smallest classes (3 nm to 700 nm) as a basis of the nucleation, Aitken, and accumulation subranges that will constitute the target variables of our study, which are compared against measurements (see Section 3 for more details). Additionally, all ten size classes are included as input variables in the ML models, along with the other simulated variables on which the downscaling is based (see Table S1)"

Additionally, this comment helped us notice that in Section 3, we had not explicitly mentioned that the SALSA bins are summed to form the three subranges (modes). We did so by adding a sentence on line 131:
"The three subranges that constitute the target variables of the study were formed by summation over the relevant bins."

As for the reviewers comment about conversions and postprocessing steps, we wanted to clarify this by adding the following sentence on line 135:
"No other conversions or postprocessing steps were performed."

2. The selected Machine Learning (ML) algorithms are often used for regression and classification tasks, but it would be valuable for the authors to clarify whether they considered alternative models specifically designed for time series prediction, such as Long Short-Term Memory (LSTM) networks or Recurrent Neural Networks (RNNs). These models are well-suited for capturing temporal dependencies and trends, which may be relevant for predicting daily Particle Number Concentration (PNC). A brief discussion or justification of

the model's selection, particularly regarding the temporal structure of the data, would strengthen the manuscript.

This is an important point, as the reviewer is correct in noting that deep learning methods such as RNNs and LSTMs can be well-suited for predicting PNC time series. However, given the broad range of methods already included in our comparison, we ultimately decided to limit the scope of this study to non-deep learning methods. LSTMs, and RNNs in general, require uninterrupted data sequences, which in our case would either necessitate some method of imputation or significantly reduce the usable data. However, as deep learning would be a promising approach for future studies, we have added a mention of this on line 548:
"Additionally, deep learning methods specialized for time series regression, such as Long Short-Term Memory NNs (Hochreiter and Schmidhuber, 1997), might further improve the downscaling performance by accounting for temporal dependencies in the observed and simulated data (if training data with no missing values were available)."

3. The feature selection procedure described in Section 4.3, based on iterative removal of features with the highest number of correlations above a threshold (red_thresh), is not a standard approach in the literature. It would be helpful if the authors could clarify the reasons for choosing this specific method over more standard approaches, such as filtering correlated pairs directly or using model-based feature importance metrics. Additionally, a brief discussion on the potential risks of this approach, such as removing features that may provide complementary information, would strengthen the methodology.

We thank the reviewer for raising these concerns and agree that it is important to justify the selected methodology. This decision to use a custom procedure was driven by the aim of having a flexible, entirely data-driven workflow, in which feature selection would also be optimized on a case-by-case basis. We were inspired by the mRMR method, which works in a very similar way. At an earlier point of the study, we compared our approach to both mRMR and a purely redundancy-based approach (referred to by the reviewer as filtering correlated pairs). Our approach generally resulted in the best models compared to the alternatives. The comparison, which was part of a master's thesis project, was considered slightly out of scope to be presented in our study and therefore, the results were not included in the manuscript.
To clarify our objectives, we have made the following modifications to the manuscript:
L207: added the sentences "Each criterion alone, however, provides only a partial view and could result either in an ineffective selection or the removal of important information. In

addition, as feature selection is still often performed manually rather than in a data-driven manner, these issues could be further exacerbated by human error."

L208: changed the sentence to "In this study, these ideas were combined to take both redundancy and relevance into account **through a data-driven approach.**"

L215: added the sentences "Our approach was inspired by the more commonly used minimum redundancy — maximum relevance (mRMR) method (Ding and Peng, 2005), which we also initially compared to both our approach and a more typical correlation-based selection procedure (not presented here). As further motivation for our approach, it is worth noting that mRMR does not offer a mechanism to adjust the relative weighting of relevance and redundancy, which could be beneficial in certain cases"

To briefly comment on the model-based feature importances as a basis of feature selection, we thought that they might cause bias towards a particular model, e.g. a Random Forest-based approach could select features that work better for Random Forests. Permutation-based approaches could be applied to any model but are also slower to compute (especially for large models) and sensitive to the presence of correlating features. Finally, the reviewer mentioned that a discussion of potential risks might be helpful. We have now done this on line 215:

"Naturally, removing any feature during selection entails a trade-off: the model loses some amount of information, even if that information appears redundant or insignificant in the training data. While feature selection is based on observed correlations and redundancy, it is still possible that a removed feature could improve predictions in future scenarios—particularly for out-of-distribution data. To mitigate this risk, we combined feature selection and model-specific hyperparameter optimization using cross-validation on a holdout validation set, allowing the process to account for generalization performance. Additionally, as a robustness check, we repeated the model training using all available input features (i.e., without applying feature selection), to compare performance and ensure that potentially valuable information was not systematically excluded."

4. While the authors provide an extensive description of the Bayesian Optimization (BO) framework, the methodology for hyperparameter tuning (Section 4.5.1) appears excessively complex for the problem and the size of the data. The reasoning for using kernel modifications for integer and categorical hyperparameters, rather than more standard methods such as grid or random search, could be better explained. It would be helpful for the authors to explain why such a sophisticated method was necessary, and whether it led to substantial improvements in model performance. Additionally, a workflow

diagram summarizing the hyperparameter tuning process could improve clarity and reproducibility.

We agree with the reviewer that the BO framework was likely more complex than was needed for our application, and we can see from the results that, in many cases, random search performed better (possibly indicating that, in these cases, pure chance had more impact on the results than the choice of the optimization method). The reason we used Bayesian optimization in the first place was not because we thought it necessary for the problem, but because we were interested in seeing whether or not it would outperform random search. Thus, in this kind of technical paper we do not see using a potentially more complex method as a problem. Additionally, future studies might utilize more complex models or larger datasets (particularly if a single model is trained for multiple stations), in which case it might help that a case study already exists using BO. In any case, we feel that a quantitative comparison of different optimization approaches is outside the scope and intention of this study. To briefly motivate the usage of BO also in the manuscript, we have added the following sentence on line 306:

While the models and datasets of this study are relatively small and thus a complex method like BO may not lead to major computational gains, the same methodology could be applied in future studies with more computationally intensive problems. These could be, for example, downscaling of a longer or higher resolution time series, training a single model on data from a large number of sites, or using large deep learning models.

We also thank the reviewer for the suggestion of the workflow diagram. We re-considered whether we could make one that clarifies the process, as we would also like to include one, but ultimately, we think it might complicate it even further. This is because the hyperparameter tuning process involves a complicated nested loop – using BO (or random search) to explore the parameter space, training and validating the ML models for each candidate point, and repeating this all for each ML method, site, size range, and optimization approach – which is  ambiguous to visualize clearly. We hope that the other modifications we have made to the text on this topic can help clarify the process.

5. The logical flow of the results section could be improved. Currently, the downscaling performance (Section 5.1) is presented before the comparison of ML models (Section 5.2). However, it seems more logical to first compare the models, justify the selection of the best performing one, and then present the downscaling results. Reorganizing the sections accordingly would help readers follow the reasoning behind model selection and evaluation.

Both reviewers have brought up this same point. We have made changes accordingly and agree with the reviewers that this has made the manuscript clearer to understand.

**Specific comments:**

Abstract:
1. The abstract would benefit from a clearer explanation of the data used as ground truth for training and validation of the ML models. It is important to specify which datasets were used as reference for the predictions.
We have added the names of the stations and the years which the data cover. We thank the reviewer for pointing out this and other things missing from the abstract.

2. The six ML models tested should be mentioned in the abstract.
This has now also been added.

3. Line 11: It would be desirable to provide a value of the "highest fit indices" mentioned.
We decided to remove the sentence ", which achieved the highest fit indices", as we agree that, if mentioned, numerical values should be provided. We feel that giving metrics in the abstract would worsen its readability, as we would then have to also provide pre-downscaling values as comparison; also, due to the differing scales of the target variables, common metrics like MAE or RMSE would not be informative, and $\rho^2$ has not yet been defined.

4. Lines 11-13: The abstract suggests that challenges in downscaling were observed only for the nucleation mode. Could the authors clarify why these challenges were specific to this mode and not evident in the others? A more detailed explanation in the relevant section would be beneficial.
This was a good remark; the abstract text might give an unbalanced view of the challenges. We have tried to improve the text by slightly adjusting the wording, and by adding a paragraph to Section 5.2 discussing the particular difficulties of the nucleation subrange (this was brought up by both reviewers).
L6: modified to "Challenges remained **particularly** in downscaling the nucleation subrange"
L380: added "The representation of new particle formation and nucleation–sized particles is, on many occasions, not sufficient in global climate models (Williamson *et al.*, 2019). This can be due to, for instance, errors in estimating nucleation rates. As a study

by Laakso *et al*. (2022) shows, ECHAM-SALSA tends to favor partitioning of sulfuric acid to the particle phase due to nucleation over condensation, which may lead to overestimation of nucleation subrange particles. Kokkola *et al.* (2018) compared ECHAM-SALSA number size distributions to observation data, and their results revealed that at some measurement stations, ECHAM-SALSA overestimates the nucleation mode number concentrations. Furthermore, ECHAM-SALSA does not model new particle formation due to nitrates, which may cause differences between modelled and measured nucleation subrange number concentrations. The representation of nucleation-sized aerosols could be enhanced by including a volatility basis set (VBS) scheme (Donahue *et al.*, 2011), which can improve the representation of secondary organic aerosols."

Sect. 2:
1. Line 112-113: Could the authors clarify whether the layer nearest to the surface was used for the analysis? Please, specify.
The ECHAM-HAMMOZ data is mostly from the lowest model layer, i.e. the layer close to the surface. However, for some stations, the station is located at such altitude that it corresponds to the second lowest layer. The ECHAM-HAMMOZ data is postprocessed to correspond to the station altitude by interpolating the data on the vertical axis.
We modified the sentence in L181 from
"For each station, we used ECHAM-HAMMOZ data from only one ECHAM grid cell, which contained the station coordinates and altitude. "
to
"For each station, we used ECHAM-HAMMOZ data from only one ECHAM grid cell, which contained the station coordinates and altitude. The ECHAM-HAMMOZ data with a vertical dimension were interpolated to correspond to the station altitude. This was done by utilizing the CDO command line tool (Schulzweida, 2023), and by using the nearest layer to the surface as well as the second lowest layer."

Sect. 4.1:
1. The inputs to the ML models in Section 4.1 are not clearly described. Please clarify which ECHAM-HAMMOZ outputs were used, and whether the models were trained separately for each station or using combined data. This is explained later but it would be beneficial to include that information also earlier in this section.
We thank the reviewer for these suggestions that help improve the clarity of the manuscript. We have made the following addition to Section 4.1, line 156:

"The inputs are properly introduced in Section 4.2 and listed exhaustively in Table S1. The downscaling is site-specific, that is, the ML models are trained separately for each station and size subrange."

Additionally, in response to reviewer #1's comment of similar nature, we have mentioned the site-specificity of the models in the abstract on line 9:

"A separate ML model was trained for each of the sites and size ranges."

2. Line 167-168: Was k-fold cross-validation used in any stage of the analysis? Please, provide details.

To make the wording of this part clearer and more exact, we have altered it as follows:

L168: modified the sentence to "Rolling variants of **k-fold** cross-validation that retain the ordering of the data were considered..."

L170: added "Thus, k-fold cross-validation was not used in this study."

Sect. 4.2:

1. Lines 181-182: If two of the stations fall within the same ECHAM grid cell, does this mean the simulated data for these two stations is identical, while the observed data differs? If this is the case, how was this handled in the analysis?

The reviewer is correct in stating that the simulated data from two stations located in the same grid cell are identical, and that the only difference is in the observations. This is part of the reason why we wanted to include both Melpitz and Leipzig stations, as this shows how the downscaling can be done using identical inputs as long as distinct measurements are available. We hope that specifying that separate models were trained for each site (see the first comment on Section 4.1) answers the reviewer's question of how this was handled in the analysis.

2. Lines 186-187: I suggest to explain better to which variable do the authors refer with "winter-summer variability" and "spring-summer variability".

We have made the following modifications to Section 4.2 to clarify how these variables, and also the other cyclic variables, work:

L187: changed the sentence to "These were also formed cyclically **(and thus vary between -1 and 1)** to avoid a large difference between the value of the last day of the year and the first of the next year..."

L188: added the sentence: "In practice, transforming a variable into two cyclic components was done by applying sinusoidal functions (sine and cosine) to the proportion of the variable's current value relative to its maximum (for example, time_ws = cos([day number]/[days in year])."

L189: changed the sentence to "In our naming convention for all four cyclic features, the positive direction is referenced first (i.e., wind_ns = 1 means northerly wind **and time_ws = 1 means New Year**)"

3. Lines 193-198: The manuscript mentions splitting the dataset by year (one split per year). Is this a common practice in similar studies? Furthermore, how were missing values handled in the data?

Splitting data by year preserves seasonality in the data (due to e.g. emissions and boundary layer dynamics) in an easily understandable way and thus it is commonly applied in these kinds of studies. An example of this is a study by Ivatt and Evans (2020), which additionally found considerable benefits from using at least 8 months of training data, although their study differs from ours in many aspects.

Missing data was not imputed for three reasons: 1. There was not very much of it 2. The statistical methods do not depend on continuous time series 3. This would have caused another source of uncertainty in the results.

The addition made on line 135 of the manuscript as a response to the reviewer's first general comment accounts for all postprocessing steps, including imputation. Thus, we think no further clarifications are needed in the manuscript.

4. Lines 199-200: Were any other data preprocessing techniques (apart from normalization) applied? If so, please specify.

Clarified on line 200: "No other preprocessing techniques were applied."

Sect. 4.3:

1. Lines 209-211: When two variables were found to be highly correlated, which one was dropped? Additionally, is there a risk that intercorrelations among other variables led to unintended feature removal? Including a correlation matrix for all variables would help clarify and visualize the feature selection process.

We thank the reviewer for pointing out that we had not mentioned what is done when only two variables exceed the correlation threshold, as they correlate with each other and will thus both have the same count of high correlations and the same sum of correlations. In that case, we have arbitrarily removed one of them based on their order in the input data columns – the one that comes first is removed. Although simple, this seems to work well – and in case it doesn't, we have the no-selection cases as an alternative. Another, more sophisticated option might be to e.g. choose the feature with the higher relevance of the two. We have added a mention of this behavior on line 212:

"If only two highly correlated features remain and the sums are thus equal, we have removed the one that appears earlier in the column order of the input data."

As for the risks involved in feature selection, we hope that our response to the reviewer's third general question already answers this. In addition, we have improved the clarity of the section through a few modifications, as unclear phrasing and a complicated sentence structure may have made the section difficult to understand:
L209-210: modified the sentence as follows: "For each feature, the number of **high-correlation pairs (correlations larger than red_thresh) that included the feature** was counted**. T**he feature **participating in the largest number of such pairs** was **then** dropped."
L211: changed the sentence to "In case the **number of high-correlation cases** was equal for two **or more** features, **the magnitudes of the correlations were compared, and** the one with the larger sum of correlations was dropped."

The reviewer also suggested including a correlation matrix. While this would be interesting to show, the matrix would be 100 x 100 in size, and thus too large to be clearly visualized. Additionally, the correlation matrix is only half of the story, as the relevance vector (correlation with the target) also affects the selection.

Sect. 4.6:
1. In general, this section could benefit from a clearer and more concise organization of ideas. Please, revise it.
We have re-formulated the entire section to improve its clarity and conciseness. For all the changes, see the manuscript document.
One important change is that we had mistakenly written that we used Spearman's correlation, while we actually used Pearson's. As we are comparing predictions to measurements, a non-linear dependence is not wanted and should not be rated highly.

Sect. 5.1:
1. Figure 3: Please clarify in the caption and text that the figure corresponds to the test dataset.
This is an important clarification. We have modified the caption and text accordingly.
Caption of Fig. 3: changed to "Daily average particle number concentrations per subrange in 2018 **(test set)**, for all three sites."
L356: changed the sentence to "Figure 4 shows the PNC results of the most successful downscaling methods for **the test subset (2018) of** each of the eight datasets."

Sect. 5.2:

1. Figure 4: The overall performance of the ML models appears low. A comparison with previous studies or a discussion of whether a $\rho^2$ of ~3 represents a significant improvement would strengthen the interpretation of the results.

We agree that discussion of similar studies is important, especially since purely comparing prediction metrics can make our models seem weak. We have written a new paragraph starting from line 399, discussing our results relative to previous studies:

"We can compare our results to previous studies to place them in a broader context, although no directly comparable studies exist. For example, Ivatt and Evans (2020) trained an XGBoost model to improve the ozone predictions of a chemistry transport model, and achieved an improvement in Pearson's $r$ of 0.36 (from 0.48 to 0.84). This is similar, though in most cases slighty higher, to the improvement achieved by our models. In addition to a different target variable, their higher time resolution and lower spatial resolution (mean of multiple sites) complicate the comparison. XGBoost was also the most successful model in the study by Venuta $et\ al.$ (2024), which produced spatiotemporal UFP predictions (logarithm of PNC) with a daily time resolution. Their $\rho^2$ of about 0.72 was significantly higher than ours, though a direct comparison is again difficult due to the smoothing effect of the log-transform combined with data trimming they performed. Additionally, they used observational weather data instead of climate simulations to train the models. As our models can in theory be used to predict far into the future and produce non-transformed estimates of UFP concentrations, the seemingly lower metrics are still competitive, especially considering the substantial improvement over the ECHAM predictions."

2. Lines 422-423: The text suggests that the particle number size distribution in Helsinki is the sole factor affecting modeling complexity. Could the authors consider that the input data and variable selection might also contribute to this issue? Focusing solely on a single factor may oversimplify the problem.

In addition to the new paragraph on line 380 that discusses the modeling complexity of nucleation mode, it is a good idea to also mention that the size distribution is likely not the only contributing factor. We have added the following sentence at the end of the aforementioned paragraph:

"In addition to limitations in representing the nucleation mode, other input variables can also contribute to challenges in downscaling. The coarse spatial resolution of global-scale models naturally limits their ability to accurately capture processes other than just new particle formation."

It is technically possible that feature selection removed information useful for predicting the test set – information that seemed irrelevant in terms of the training and validation sets. However, this would rather be due to a lack of representativeness of one or more of the subsets than directly due to feature selection.

3. Line 425: The text mentions feature selection, but this is not clearly indicated in Tables S2–S8. Please revise.
To make it clearer to the reader how feature selection relates to hyperparameter optimization, we have modified a sentence in the tables' caption from
"The number of features after selection (controlled by the thresholds rel_thresh and red_thresh) are also reported in parentheses."
to
"Model-specific feature selection was performed during hyperparameter optimization through two additional parameters, rel_thresh and red_thresh. The number of features after selection, controlled by the threshold parameters, is also reported in parentheses."

4. Lines 423-432: It is unclear how the number of features selected for each model were determined. Please clarify.
To clarify this and make the relevant information easier to find, we modified a parenthetical expression on line 425 from
"(see Tables S2–S8)"
to
"(see Section 4.3 for a description of feature selection, and Tables S2–S8 for optimization results)"

Sect. 5.3:
1. The section would benefit from a more structured presentation, perhaps summarizing the main conclusions at the end. Clarifying how the feature selection approach interacts with the optimization process and providing a clearer link to the main results of the study would improve the clarity and relevance of this discussion.
We thank the reviewer for pointing out that the structure of this section was not as clear as we had thought. We have restructured the section so that 1. The first paragraph discusses the colors of the table and their interpretations (which were originally split between two paragraphs). 2. The second paragraph compares BO and RS and discusses potential shortcomings. 3. The third paragraph addresses feature selection, and 4. Some conclusions are made in the fourth paragraph, which has not changed since the previous version.

Additionally, to clarify the feature selection / optimization process to the reader and direct them to the relevant sections of the manuscript, we have added the following sentence on line 447:

"Simultaneously with the model hyperparameters, feature selection was also optimized through two threshold parameters for redundancy and relevance (see Sections 4.3 and 4.5.2)."

This also helps the structure of the section, making it clear that the third paragraph focuses particularly on feature selection.

2. Lines 439-441: Could the authors clarify whether these hyperparameters were intended to be the final optimized values?

We assume that by "these hyperparameters", the reviewer refers to the Bayesian optimizer parameters kappa, alpha, and nu mentioned on line 440. As mentioned in Section 4.5.2, a few alternative parameter values were compared, but they were not explicitly optimized. It should not be necessary to optimize an optimizer.

We also added the following clarification to the caption of Figure 3 (now Figure 4) to better connect the two figures:

"Measurements are shown in blue, ECHAM-HAMMOZ outputs in green, and the results of downscaling by the best model for each dataset **(i.e., the bolded cells in Figure 3)** in **purple**"

Sect. 5.4:

1. Line 496-498: It would be helpful if the authors could specify which features were selected for each model.

For the SVM model of Helsinki's accumulation subrange, all 100 features were used. As for the NN2 model, we have added Table S9 to the Supplementary material to list the 18 features used by it. We have not listed features for other models, as even just showing the best models' features for all datasets would result in 56 of such lists, which would take up too much space. The top 9 features of each model are still shown in the SHAP plots in the Supplementary material.

L498: modified the sentence to "It is interesting to note that NN2 has almost the same $\rho^2$ as SVM, despite using far fewer features (18 and 100, respectively**; see Table S9**)."

2. Line 504-508: While the authors use SHAP analysis to indicate feature importance, it is important to note that SHAP values reflect each feature's contribution to the model's predictions, not necessarily a causal relationship with the predicted quantity (PNC in this case). For example, the importance of sea salt may result from correlations with other

variables (e.g., num_2a6, num_2a7, WAT_2a6, WAT_2a7) more directly linked to PNC. The authors should clarify this distinction and discuss the implications of such proxy features in interpreting the model outputs.

We have modified this part to further clarify that SHAP explains models, not "real" cause-and-effect relationships. This is an important distinction, and was the reason why we mentioned the sea salt example in the first place, though we did not explicitly state our intentions.

L504: changed to "Likely, some (or even most) of the features shown in the SHAP plots are only deemed important **in terms of their contribution to the downscaling** because they correlate with some physically relevant quantity, …"

L508: added sentence "Hence, when employing SHAP values to assess feature importance, it is important to note that SHAP explains how specific models operate and is not to be interpreted as a tool for causal inference of physical systems."

Supplementary Material:

1. The hyperparameter values reported in Tables S2-S8 raise questions. For example, in Table S4, learning rates such as 0.547 (Leipzig Acc) or 0.189 (Helsinki Nuc) are atypically high compared to standard practices in XGBoost modeling, where learning rates are typically in the range of 0.01 to 0.1. Similarly, the large values for regularization parameters (e.g., reg_alpha = 708 for Leipzig Acc) seem unusual and potentially indicative of overfitting or instability. The authors should discuss the implications of these unusual values and whether they align with expected behavior in aerosol-climate model downscaling.

We sincerely appreciate the reviewer's detailed feedback on the manuscript as well as the Supplementary material. The XGBoost for Helsinki Nuc is a very weak model in the first place ($\rho^2$ = 0.047), so it's not unexpected that its hyperparameters can look atypical (notice also that only 5 features are used by the model). In the case of the XGBoost for Leipzig Acc, the model seems to perform quite well, so we hypothesize that the high learning rate might be connected to the smaller size of the trees in this model (small max_depth and large min_child_weight), and/or the strong regularization due to the high reg_alpha. We have made the following addition to the manuscript to connect the hyperparameters of the weak XGBoost model to its performance:

L406: changed the sentence to "It only failed in the nucleation dataset of Helsinki, where it had a lower $\rho^2$ than any other model **(see also Table S4 for hyperparameters, some of which are atypical).**"

**Minor comments:**

Lines 43-44: This sentence looks incomplete.

We have moved the word "however" from the end of the sentence to the start.

Line 91: "were" instead of "was".

In this case, "was" refers to the word "selection", which is singular.

Line 145-147: Please, revise this sentence.

We have changed the wording to reduce the repeated use of the words "Leipzig" and "station". We also made similar changes to the other paragraphs of this section.

Line 164: "a" instead of "an".

According to grammar guides (e.g. Merriam-Webster), "an" is correct here. However, this comment helped us notice that we had written "a ML" on line 267. This was corrected to "an ML".

Line 393-394: Where is this comparison?

This was done for a master's thesis project, and is outside the scope of our study. Therefore, we have not presented this comparison in the manuscript, and have clarified that in the text:

L393: modified to "The eight datasets were further examined through statistical tests comparing the means of the training, validation and testing subsets of the measured PNC **(not shown).**"

Caption of Figure 3: Is it orange?

Corrected to purple. This was a leftover from before we improved the accessibility of the figures.

References:

Bergman, T. ,Kerminen, V.-M., Korhonen, H., Lehtinen, K. J., Makkonen, R., Arola, A., Mielonen, T., Romakkaniemi, S., Kulmala, M., and Kokkola, H.: Evaluation of the sectional aerosol microphysics module SALSA implementation in ECHAM5-HAM aerosol-climate model., Geosci. Model Dev., 5, 845–868, https://doi.org/10.5194/gmd-5-845-2012, 2012.

Ding, C. and Peng, H.: Minimum redundancy feature selection from microarray gene expression data, Journal of Bioinformatics and Computational Biology, 3, 185–205, https://doi.org/10.1142/s0219720005001004, 2005

Donahue, N. M., Epstein, S. A., Pandis, S. N., and Robinson, A. L.: A two-dimensional volatility basis set: 1. organic-aerosol mixing thermodynamics, Atmos. Chem. Phys., 11, 3303–3318, https://doi.org/10.5194/acp-11-3303-2011, 2011.

Hochreiter, S. and Schmidhuber, J.: Long Short-Term Memory, Neural Computation, 9, 1735–1780, https://doi.org/10.1162/neco.1997.9.8.1735, 1997

Ivatt, P. D. and Evans, M. J.: Improving the prediction of an atmospheric chemistry transport model using gradient-boosted regression trees, Atmos. Chem. Phys., 20, 8063–8082, https://doi.org/10.5194/acp-20-8063-2020, 2020.

Kokkola, H., Kühn, T., Laakso, A., Bergman, T., Lehtinen, K. E. J., Mielonen, T., Arola, A., Stadtler, S., Korhonen, H., Ferrachat, S., Lohmann, U., Neubauer, D., Tegen, I., Siegenthaler-Le Drian, C., Schultz, M. G., Bey, I., Stier, P., Daskalakis, N., Heald, C. L., and Romakkaniemi, S.: SALSA2.0: The sectional aerosol module of the aerosol–chemistry–climate model ECHAM6.3.0-HAM2.3-MOZ1.0, Geosci. Model Dev., 11, 3833–3863, https://doi.org/10.5194/gmd-11-3833-2018, 2018.

Laakso, A., Niemeier, U., Visioni, D., Tilmes, S., and Kokkola, H.: Dependency of the impacts of geoengineering on the stratospheric sulfur injection strategy – Part 1: Intercomparison of modal and sectional aerosol modules, Atmos. Chem. Phys., 22, 93–118, https://doi.org/10.5194/acp-22-93-2022, 2022.

Schulzweida, U.: CDO User Guide, https://doi.org/10.5281/zenodo.10020800, 2023

Venuta, A., Lloyd, M., Ganji, A., Xu, J., Simon, L., Zhang, M., Saeedi, M., Yamanouchi, S., Lavigne, E., Hatzopoulou, M., and Weichenthal, S.: Predicting within-city spatiotemporal variations in daily median outdoor ultrafine particle number concentrations and size in Montreal and Toronto, Canada, Environmental Epidemiology, 8, e323, https://doi.org/10.1097/EE9.0000000000000323

Williamson, C.J., Kupc, A., Axisa, D. *et al:* A large source of cloud condensation nuclei from new particle formation in the tropics, *Nature,* **574**, 399–403, https://doi.org/10.1038/s41586-019-1638-9, 2019.